# Plasma Cytokine Levels and Cytokine Genetic Polymorphisms in Patients with Metastatic Breast Cancer Receiving High-Dose Chemotherapy

Robert Lafrenie [1,2,3,*] , Mary Bewick [1], Carly Buckner [1,2] and Michael Conlon [1]

1   Health Sciences North Research Institute, Sudbury, ON P3E 2H3, Canada
2   Program in Biomolecular Sciences, Laurentian University, Sudbury, ON P3E 2C6, Canada
3   Department of Medical Sciences, Northern Ontario School of Medicine, Sudbury, ON P3E 2C6, Canada
*   Correspondence: rlafrenie@laurentian.ca

**Abstract:** Differences in the baseline levels of serum cytokines or in single-nucleotide polymorphisms (SNPs) in cytokine genes may be useful to predict outcomes for patients being treated for metastatic breast cancer. We have measured the plasma levels and characterized individual SNPs for IL-1RA, IL-1β, IL-2, IL-6 and TNFα in 130 patients with metastatic breast cancer treated with high-dose chemotherapy. Patients were treated with high-dose cyclophosphamide (Group 1, 74 patients) or high-dose paclitaxel-containing regimens (Group 2, 56 patients). A high plasma level of IL-1RA and a SNP in the IL-1RA gene indicated a better prognosis for patients in Group 1 (but not Group 2). However, the level of plasma IL-1RA did not correlate with the SNP genotype. A high plasma level of IL-6 or TNFα indicated a poorer outcome for patients in Group 1 although the SNP genotypes for the IL-6 and TNFα SNPs were not associated with differences in outcome. The plasma levels of IL-1β and IL-2 and the genotype of the IL-1β SNPs did not indicate differences in outcome. Although, individually, plasma levels of cytokine or "risk" SNP genotypes may not indicate outcome, in combination there was an increased trend to predict outcome for patients treated with high-dose cyclophosphamide but not high-dose paclitaxel. These results suggest that the immune cytokines may be useful as prognostic biomarkers in the treatment of patients with metastatic breast cancer treated with different types of chemotherapy.

**Keywords:** breast cancer; interleukin-1 receptor antagonist; cytokines; chemotherapy; prognosis





## 1. Introduction

Host-derived biomarkers can be used to stratify prognosis for patients with breast cancer. Immune mechanisms may play a critical role in cancer prognosis and successful treatment [1,2]. The baseline levels of several cytokines and other immune-related proteins in tumor tissue or plasma have been associated with the risk of breast cancer and with the prognosis of patients diagnosed with breast cancer undergoing treatment [3–5]. For example, the interleukin-1 family of cytokines, which includes IL-1α and IL-1β, are implicated in carcinogenesis [6] and high serum levels correlate with a poorer prognosis. Cytokines such as IL-1 and TNFα are associated with an inflammatory response and inflammation has been associated with a poorer clinical outcome for patients with cancer [7]. The anti-inflammatory cytokine IL-1RA inhibits the effects of IL-1β and is associated with improved prognosis [8,9]. Patients with breast cancer have higher levels of serum IL-6 and TNFα than patients without cancer and high levels of IL-6 [10,11] and TNFα [12] correlate with a poorer outcome [13]. Higher levels of IL-8 [14], IL-10 [15], and TGFβ have also been correlated with an improved prognosis for women with breast cancer [16,17]. In patients with HER-2-positive breast cancer, IL-2, TNFα [18], and IL-6 [19] levels were elevated and associated with a poorer clinical outcome [20]. Although the level of several different individual cytokines and other immune molecules have been associated with

clinical outcomes for patients with cancer, the evaluation of multiple related blood or tissue biomarkers is more reliable for indicating prognosis [4,21,22].

Specific differences in the DNA sequence of several cytokine genes have been associated with cancer risk or prognosis [23,24]. For example, single-nucleotide polymorphisms (SNPs) in the IL-1β gene have been linked to an increased risk of cancer, including breast cancer [25,26], and with clinical outcome [27,28]. SNPs in the IL-1RA gene are also associated with clinical outcome for various cancers and have a stronger effect in combination with risk IL-1α or IL-1β SNPs [7,29,30]. SNPs in the IL-2, IL-6, and TNFα genes have also been associated with prognosis for patients with breast cancer [31–35]. These studies suggest that the best risk assessments require contributions from multiple cytokine SNPs although a single cytokine SNP can indicate clinical outcome [24].

Some polymorphisms in cytokine genes, such as IL-1β (rs1143634) [36], IL-1RA (rs4251961) [37,38], IL-6 (G/C 174 rs1800795) [39,40], and TNFα (G/A 308 rs1800629) [41], are associated with the level of protein expression which is thought to contribute to the mechanisms of action; however, other polymorphisms do not correlate with protein expression or function and may be linked to prognosis through alternate mechanisms or by linkage with other functional polymorphisms [42].

In this study, we examined the association of the IL-1β, IL-1RA, IL-6, IL-2, and TNFα biomarkers with cancer progression and survival outcome for patients with metastatic breast cancer treated with high-dose chemotherapy including either cyclophosphamide or paclitaxel. In addition, the protein expression levels in plasma and SNP genotypes of the same cytokine were evaluated as biomarkers associated with clinical outcome.

## 2. Materials and Methods

### 2.1. Patient Population

The plasma levels of cytokines and cytokine genetic polymorphisms were examined in women with metastatic breast cancer treated with high-dose chemotherapy (HDC) and autologous stem cell transplantation (ASCT). Patient characteristics and HDC treatment regimens are listed in Table 1. Patients were selected from 130 women enrolled in clinical trials of HDC with ASCT at Sudbury Regional Hospital between 1991 and 1997, and details of the patients' response to treatment have been published previously [43,44]. The clinical trials and study were approved by the Research Ethics Board, Sudbury Regional Hospital, Laurentian Site, Sudbury, Ontario and informed signed consent was obtained from all patients. The patients in this study were separated into 2 groups based on chemotherapy regimen. There were 74 patients in Group 1 who were enrolled in 4 phase II clinical trials between 1991 and 1994 to study HDC treatment comprised of high-dose cyclophosphamide, mitoxantrone, and vinblastine or carboplatin [45,46]. There were 56 patients in Group 2 who were enrolled in a phase I/II clinical trial between 1994 and 1997 and treated with HDC consisting of cyclophosphamice, mitoxantrone, and high-dose paclitaxel [44,47]. Eligible patients were histologically diagnosed with metastatic breast cancer (Stage IV), had a Karnofsky performance status of ≥60%, did not have CNS metastases, and had not received chemotherapy for metastatic breast cancer or adjuvant chemotherapy for at least 6 months prior to enrolment. Some information, such as estrogen receptor, progesterone receptor, and HER-2 status, was available for a subset of patients.

**Table 1.** Clinical characteristics of patients with metastatic breast cancer treated using high-dose chemotherapy including cyclophosphamide (Group 1) or paclitaxel (Group 2).

| Clinical Characteristic | Total | Group 1 | Group 2 |
|---|---|---|---|
| | (N = 130) | (N = 74) | (N = 56) |
| Age | | | |
| <40 | 32 | 21 | 11 |
| 40–49 | 66 | 39 | 27 |

**Table 1.** *Cont.*

| Clinical Characteristic | Total | Group 1 | Group 2 |
|---|---|---|---|
| | (N = 130) | (N = 74) | (N = 56) |
| 50–59 | 32 | 14 | 18 |
| ER positive (n = 113) | 66 | 39 | 27 |
| PR positive (n = 108) | 57 | 37 | 20 |
| HER-2 positive (n = 117) | 54 | 29 | 25 |
| Number of Metastatic Sites | | | |
| ID, 1 | 75 | 48 | 27 |
| ≥2 | 55 | 26 | 29 |
| Metastatic sites | | | |
| Bone | 65 | 36 | 29 |
| Lung | 41 | 20 | 21 |
| Lymph Node | 42 | 21 | 21 |
| Liver | 21 | 11 | 10 |
| Other | 26 | 14 | 12 |
| HDC regimen | | | |
| Mitox, Cyclo, Vin | 35 | 35 | |
| Mitox, Cyclo, Carbo | 29 | 29 | |
| Mitox, Cyclo, Paclitaxel | 56 | | 56 |
| Thiotepa, Cyclo, Carbo | 8 | 8 | |
| Mitox, Cyclo | 2 | 2 | |

### 2.2. Cytokine Quantitation in Plasma Samples

Plasma samples were obtained from peripheral blood samples (5 mL, heparin collection tube) drawn on day 1 or 2 of apheresis collection (before HDC treatment). Plasma was obtained by centrifugation of the peripheral blood samples at $300\times g$ for 10 min and aliquots stored at $-80\,^{\circ}$C. IL-1RA, IL-1β, TNFα, IL-6, and IL-2 plasma levels were quantitated using commercially available Parameter human ELISA kits from R & D Diagnostics, (Minneapolis, MN, USA). Cutpoints for plasma marker status were determined by dividing the patients into 10 groups (quantiles) based on increasing marker concentration. Each quantile was then tested as a cutpoint in subsequent Kaplan–Meier estimates [48] of the overall survivorship for breast-cancer-specific survival (BCSS). The maximum separation of the survivorship curves was considered to be the "optimum cutpoint" (Supplementary Table S1) and patients whose markers were greater than or equal to the cutpoint were classified as positive. For a subset of 51 patients from Group 1, samples during and after chemotherapy were available and changes in cytokine levels during treatment were also reported separately for this group.

### 2.3. Analysis of SNPs

DNA was extracted from cryopreserved, apheresis blood product or peripheral blood using the DNA Blood MiniKit (Qiagen, Mississauga, ON, Canada). A candidate approach was used to select SNPs in the IL-1RA (rs579543; rs4251961), IL-1β (rs16944; rs1143634), IL-6 (rs1800795), and TNFα (rs1800629) genes that were previously reported to be associated with chemotherapeutic sensitivity and cancer risk, progression, survival, and/or associated with protein expression [29,35,37,38,40]. Referenced TaqMan® assays, obtained from Applied Biosystems (Foster City, CA, USA), consist of two PCR amplification primers and two allele-specific fluorescent probes as described (Supplementary Table S2). Genotyping was conducted using the ABI PRISM® 7900HT Sequence Detection System. For quality-control purposes, random samples were repeated for each SNP (n = 10% of all samples genotyped). Assignment of genotypes was performed independently by two investigators blinded to the

survival endpoints. Deviations from Hardy–Weinberg equilibrium for each SNP genotype were assessed using the Pearson $\chi^2$ test.

*2.4. Statistical Methods*

Survival curves were generated using the Kaplan–Meier product limit estimate of the survivorship function [48]. Two end-points, progression-free survival (PFS) and breast-cancer-specific survival (BCSS), were used. PFS was defined as the time (months) from study enrolment until documented progression of metastatic disease or censorship. BCSS was defined as the time (months) from study enrolment until death from metastatic disease or censorship. Equality of survivorship functions' status (positive or negative) was tested for both end points using the log-rank test for the entire cohort or for patients in Group 1 or Group 2.

The Cox proportional hazard regression model defined hazard ratios (HR) and 95% confidence intervals (CI) [49]. HR and 95% CI were calculated for each independent variable for both PFS and BCSS for each group. The association between protein marker status (for IL-1RA, IL-1β, IL-2, IL-6, and TNFα) and various clinicopathological characteristics was investigated using either the Pearson's test for association (2 × n tables) or Fisher's exact test (2 × 2 tables).

**3. Results**

*3.1. Patient Characteristics*

The summary of patient characteristics including age, hormone receptor status, metastasis, and treatment regimen are shown in Table 1. These data include results from two groups of patients with very similar clinical characteristics but with differences in chemotherapy treatment; patients in Group 1 were treated with high-dose cyclophosphamide-containing chemotherapy and patients in Group 2 were treated with high-dose paclitaxel-containing chemotherapy. There were no significant differences in clinicopathologic characteristics between the groups. During the follow-up period of 140 months, disease progression and death due to metastatic breast cancer (BCSS) occurred for all of the included patients. The time at risk for disease progression (PFS) ranged from 0.9 months to 136.2 months. There were no significant differences in PFS or BCSS in the different groups as shown by the Kaplan–Meier estimates (Figure 1A,B).

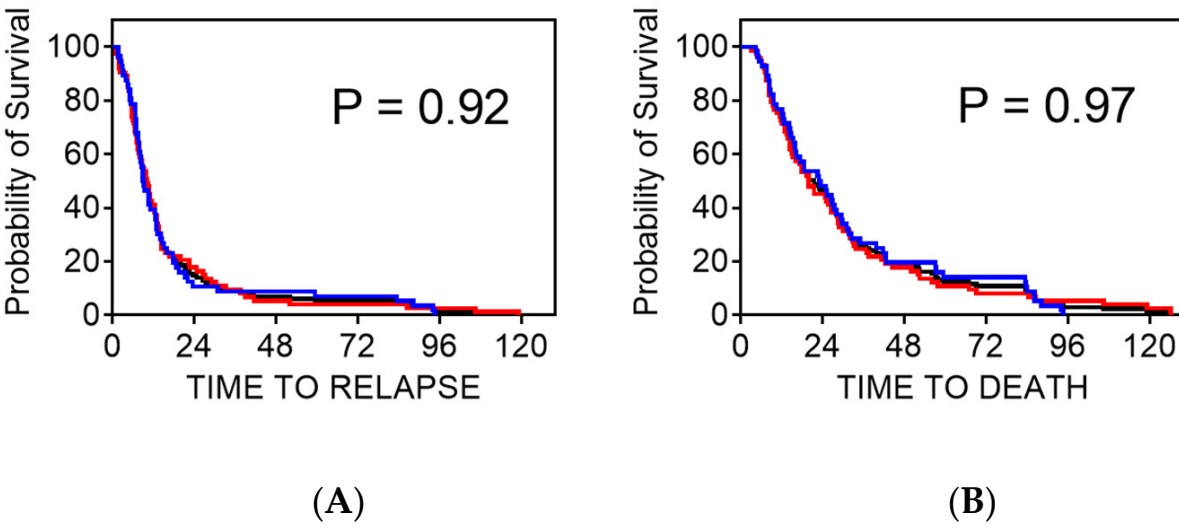

**(A)** **(B)**

**Figure 1.** (**A**,**B**) Kaplan–Meier survival curves for progression-free and breast-cancer-specific outcome for patients treated with high-dose chemotherapy including cyclophosphamide (Group 1, blue line) or paclitaxel (Group 2, red line).

### 3.2. Plasma Biomarkers

The plasma levels of IL-1RA, IL-1β, TNFα, IL-6, and IL-2 were determined for a subset of patients. Cutpoints for positivity were determined by quantile analysis of the total population of patients, and correlations between cytokine status and clinical outcome determined for the total cohort and for patients in Group 1 and Group 2. The median IL-1RA concentration for all of the patients (n = 105) was 379 ng/mL (range 0–5842 ng/mL) and the cutpoint for BCSS was 300 ng/mL (Supplementary Table S1). Patients with a high level of IL-1RA (>300 ng/mL) showed a median survival of 24.9 months compared to 16.3 months for patients with a low level of IL-1RA (HR = 2.08; 95% CI = 1.27–3.40) (Table 2). There were significant differences in PFS ($p$ = 0.049) and BCCS ($p$ = 0.0036) for the total cohort as determined by log rank analysis. Patients from Group 1 with a high IL-1RA level showed a significantly longer median survival compared to patients with a low IL-1RA level (25.1 vs. 14.4 months, HR = 2.08; 95% CI = 1.14–3.78, $p$ = 0.017) while patients from Group 2 did not show significant differences in BCCS or PFS depending on IL-1RA status.

The median IL-1β concentration for the total cohort (n = 113) was 2.24 ng/mL (range 0–402.7 ng/mL) and the cutpoint for BCSS was 3 ng/mL. Log-rank analysis showed no significant differences in PFS or BCSS between patients with a high or low IL-1β level for the total cohort or for patients in Group 1 or Group 2 (Table 2).

The median level of plasma TNFα was 16.97 ng/mL (range 0–821.1 ng/mL) for the total cohort (n = 113) and the cutpoint was determined to be 60 ng/mL. Patients with a high TNFα level showed a significantly shorter BCSS than patients with a low TNFα level (15.6 vs. 23.3 months, HR = 0.63; 95% CI = 0.40–1.00). Patients from Group 1 with a high TNFα level also had a poorer outcome than patients with a low level of TNFα (14.3 vs. 19.9 months; HR = 0.63; 95% CI = 0.48–0.92) as confirmed by log-rank analysis for BCSS ($p$ = 0.028) and PFS ($p$ = 0.007) (Table 2). The TNFα levels were not prognostic in patients from Group 2.

The median level of plasma IL-6 was 10.5 ng/mL (range 0–409.8 ng/mL) for the total cohort (n = 94) and the cutpoint was 18 ng/mL. Patients with a high IL-6 level showed no difference in median survival compared patients with a low IL-6 level for the total group. However, patients from Group 1 with a high IL-6 level showed a shorter median survival than patients with a low IL-6 level (13.4 vs. 19.9 months; HR = 0.65; 95% CI = 0.43–0.84) as confirmed by log-rank analysis for BCSS ($p$ = 0.013) and PFS ($p$ = 0.007). The IL-6 levels did not indicate prognosis for patients from Group 2 (Table 2).

The median IL-2 concentration for the total patient group (n = 101) was 0.2 ng/mL (range 0–927.8 ng/mL) and the cutpoint was 3 ng/mL. However, log-rank analysis showed no significant differences in PFS and BCSS between patients with high and low IL-2 levels for the total cohort or for patients in Group 1 or Group 2.

For each patient, the plasma levels of the different cytokines were correlated to one another. For example, the level of TNFα was shown to directly correlate to the level of IL-1RA (Pearson correlation, R = 0.688, $p \leq$ 0.0001), IL-1β (R = 0.784, $p \leq$ 0.0001), IL-6 (R = 0.832, $p \leq$ 0.0001), and IL-2 (R = 0.611, $p \leq$ 0.0001) (Table 3). This shows that patients with elevated levels of plasma TNFα also expressed elevated levels of the other cytokines.

**Table 2.** Plasma cytokine levels and clinical outcomes for patients treated with high-dose chemotherapy including cyclophosphamide (Group 1) or paclitaxel (Group 2).

| Breast Cancer Specific Survival | | | | | | | | | | | | | | | |
| --- | --- | --- | --- | --- | --- | --- | --- | --- | --- | --- | --- | --- | --- | --- | --- |
| | | Total | | | | | Group 1 | | | | | Group 2 | | | |
| Plasma Marker | N | Median Survival (Months) | Hazard Ratio (95% CI) | $X^2$ | *p* | N | Median Survival (Months) | Hazard Ratio (95% CI) | $X^2$ | *p* | N | Median Survival (Months) | Hazard Ratio (95% CI) | $X^2$ | *p* |
| IL-RA1 | 105 | 16.3 vs. 24.9 | 2.08 (1.27–3.40) | 8.5 | 0.0036 | 60 | 14.4 vs. 25.1 | 2.08 (1.14–3.78) | 5.74 | 0.017 | 45 | 18.5 vs. 24.1 | 1.91 (0.781–4.66) | 2.1 | 0.16 |
| IL-1 | 113 | 19.9 vs. 19.6 | 0.87 (0.59–1.28) | 0.5 | 0.48 | 60 | 17.7 vs. 18.8 | 1.04 (0.26–1.75) | 0.2 | 0.88 | 53 | 24.1 vs. 18.6 | 0.67 (0.374–1.22) | 1.7 | 0.19 |
| TNF | 113 | 23.3 vs. 15.6 | 0.63 (0.40–1.00) | 3.8 | 0.05 | 60 | 19.9 vs. 14.3 | 0.65 (0.48–0.92) | 4.85 | 0.028 | 53 | 25.1 vs. 16.0 | 0.82 (0.422–1.61) | 0.32 | 0.57 |
| IL-6 | 94 | 21.4 vs. 17.4 | 0.72 (0.44–1.18) | 1.71 | 0.19 | 54 | 19.9 vs. 13.4 | 0.60 (0.43–0.84) | 6.12 | 0.013 | 40 | 22.9 vs. 29.5 | 1.31 (0.672–2.56) | 0.63 | 0.43 |
| IL-2 | 101 | 22.9 vs. 15.9 | 0872 (0.58–1.32) | 0.41 | 0.52 | 59 | 19.2 vs. 16.8 | 1.10 (0.58–1.69) | 0.0003 | 0.98 | 42 | 27.5 vs. 16.0 | 0.56 (0.283–1.12) | 2.71 | 0.1 |
| Progression free survival | | | | | | | | | | | | | | | |
| IL-RA1 | 105 | 8.5 vs. 10.7 | 1.59 (1.01–2.51) | 3.9 | 0.049 | 60 | 8.5 vs. 10.7 | 1.56 (0.89–2.75) | 2.5 | 0.12 | 45 | 8.5 vs. 10.6 | 1.93 (0.74–4.71) | 2.1 | 0.15 |
| IL-1 | 113 | 9.8 vs. 9.3 | 0.81 (0.55–1.19) | 1.2 | 0.28 | 60 | 10.3 vs. 8.3 | 0.89 (0.59–1.56) | 0.19 | 0.67 | 53 | 9.7 vs. 9.3 | 0.82 (0.46–1.44) | 0.48 | 0.48 |
| TNF | 113 | 10.5 vs. 7.2 | 0.52 (0.32–0.83) | 7.3 | 0.0068 | 60 | 10.7 vs. 6.9 | 0.29 (0.14–0.59) | 11.5 | 0.007 | 53 | 10.6 vs. 9.2 | 0.84 (0.43–1.62) | 0.28 | 0.6 |
| IL-6 | 94 | 10.6 vs. 7.2 | 0.60 (0.36–1.01) | 3.8 | 0.052 | 54 | 10.7 vs. 4.7 | 0.29 (0.19–0.57) | 11.5 | 0.007 | 40 | 11.6 vs. 10.1 | 1.00 (0.48–2.08) | 5E-05 | 0.99 |
| IL-2 | 101 | 10.5 vs. 8.9 | 0.81 (0.54–1.23) | 0.97 | 0.32 | 59 | 10.2 vs. 7.4 | 0.77 (0.44–1.33) | 0.82 | 0.25 | 32 | 10.6 vs. 9.0 | 0.83 (0.43–1.58) | 0.33 | 0.57 |

Text in red indicates the marker is a significant indicator of different prognosis.

**Table 3.** Correlations among plasma cytokine levels for the total group of patients.

|  | IL-1RA | IL-1β | IL-2 | IL-6 |
|---|---|---|---|---|
| TNFα | 0.69 (<0.0001) | 0.79 (<0.0001) | 0.61 (<0.0001) | 0.83 (<0.0001) |
| IL-1RA |  | 0.68 (<0.0001) | 0.78 (<0.0001) | 0.65 (<0.0001) |
| IL-1β |  |  | 0.74 (<0.0001) | 0.88 (<0.0001) |
| IL-2 |  |  |  | 0.64 (<0.0001) |

The changes in cytokine levels during treatment were correlated to clinical outcome for a group of 51 patients in Group 1 where plasma cytokine concentrations were available before treatment, during chemotherapy, and after treatment. In this subset of patients, there was no significant change in the levels of IL-1β, IL-6, or TNFα in response to treatment with HDC (comparing before, during treatment, and after treatment levels) (Figure 2). However, patients that showed a decrease in IL-2 levels between the first sample and samples collected during treatment ($p = 0.04$) or between the first sample and the samples collected after treatment ($p = 0.0023$) correlated with an improved BCSS. In addition, patients that showed a decrease in IL-1β levels in samples collected between treatments and after treatment ($p = 0.05$) showed an improved BCSS.

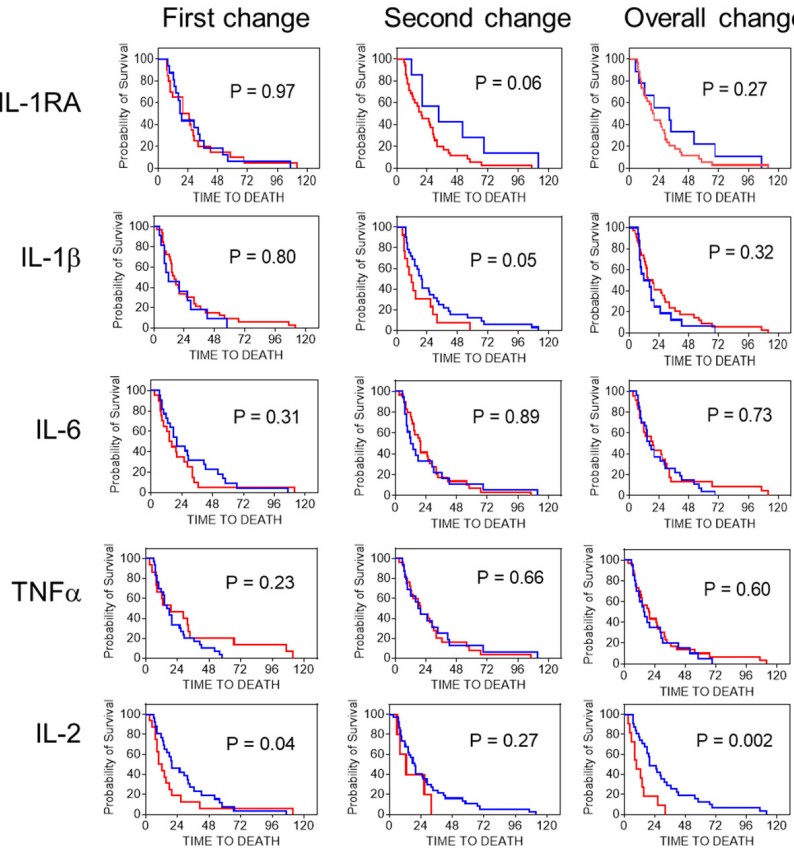

**Figure 2.** The effect of changes in cytokine levels during treatment on clinical outcome for patients treated with high-dose chemotherapy. The levels of each cytokine were compared from before treatment and mid treatment (first change), from mid-treatment to follow up (second change) or between before treatment and follow up (overall change) and patients that showed no change or increase in levels (blue line) were compared to patients that showed a decrease in levels (red line). Kaplan–Meier survival curves were created and log-rank statistics were used to evaluate any significant differences in outcome.

### 3.3. Genotypic Frequencies of Polymorphisms

The genotypic frequencies for each polymorphism are shown in Table 4. They are not significantly different from what would be expected if the population was in Hardy–Weinberg equilibrium and are similar to the frequencies reported in the NCBI SNP500 database. For each SNP, analyses were carried out for each of the genotypes separately and also for the heterozygous genotype grouped with a homozygous genotype with a similar median BCSS. Hazard ratios for BCSS and PFS are shown in Table 4. For the IL-1RA-SNP-01 (rs579543) (Supplementary Table S2), patients with the CT genotype had a median BCCS of 26.5 months, while patients with the CC genotype had a median BCSS of 15.3 months (HR = 1.6; 95% CI = 1.1–2.4, $p$ = 0.015). Patients with the CT genotype also had a longer PFS than patients with the CC genotype (10.9 vs. 8.5 months; HR = 1.5, 95% CI = 1.0–2.1, $p$ = 0.05). Patients with the CC genotype had a better outcome than patients with the combined CT + TT genotype (HR = 1.5; 95% CI = 1.0–2.1, for BCSS and HR = 1.3; 95% CI = 0.9–2.1 for PFS). Patients in Group 1 with the CT IL-1RA-SNP-01 also had a better BCSS than patients with the CC genotype (28.5 vs. 14.3 months; HR = 2.2; 95% CI = 1.3–3.8, $p$ = 0.004). For patients with the CC genotype versus patients with the combined CT + TT genotypes, the HR was 1.9 (95% CI = 1.1–3.2, $p$ = 0.019) for BCSS.

The TNF$\alpha$-SNP (rs1800629) genotype did not indicate the outcome for BCSS but patients with the GG genotype had a shorter PFS than patients with the AG genotype (8.2 vs. 10.5 months; HR = 0.62, 95% CI = 0.41–0.98, $p$ = 0.04) and with the combined AG + AA genotype (8.2 vs. 10.4 months; HR = 0.65, 95% CI = 0.43–1.0, $p$ = 0.05).

For the IL1RA-SNP-02 (rs4251961), IL-1$\beta$-SNP-01 (rs16944), IL-1$\beta$-SNP-02 (rs1143634), and IL-6-SNP (rs1800795), analyses were performed for each of the genotypes separately and also for the heterozygous genotype grouped with a homozygous genotype with similar median BCSS. This analysis showed that there were no significant differences for BCSS or PFS based on the genotype or haplotype for these polymorphisms in this patient population.

### 3.4. Comparison of Plasma Markers and Genetic Polymorphisms

A comparison of the Kaplan–Meier survival curves for BCSS for the plasma cytokine levels and cytokine SNPs are shown for the total cohort and patients in Group 1 and Group 2 (Figure 3). For IL-1RA, high plasma levels of the IL-1RA protein were associated with an improved prognosis for patients in the total population and in Group 1 but not in Group 2. Similarly, patients that expressed the CC genotype for the IL-1RA-SNP-01 had an improved prognosis compared to patients with the TT genotype for the total cohort and for patients in Group 1 but not for patients in Group 2. However, the IL-1RA-SNP-02 genotypes did not differentiate the prognosis of patients in any of the groups. There was no correlation between the plasma levels of the IL-1RA protein and either the IL-1RA-SNP-01 (Spearman's correlation, $R^2$ = 0.59, $p$ = 0.22) or the IL-1RA-SNP-02 ($R^2$ = 0.58, $p$ = 0.22) genotypes in the set of 92 patients where data were available for all measures.

For IL-1$\beta$, the plasma levels of the IL-1$\beta$ protein and the genotypes of the IL-1$\beta$-SNP-01 or IL-1$\beta$-SNP-02 were not associated with prognosis in these patients. However, the level of IL-1$\beta$ protein was shown to correlate with the IL-1$\beta$-SNP-01 ($R^2$ = 0.9; $p$ = 0.014) genotype but not with the IL-1$\beta$-SNP-02 genotype ($R^2$ = 0.97; $p$ = 0.12).

The plasma level of IL-6 correlated with outcome for patients in Group 1 but not for the total cohort or for patients in Group 2. The genotype of the IL-6 SNP did not describe a difference in BCSS for the patients and there was no correlation between the level of plasma IL-6 and the IL-6-SNP.

For TNF$\alpha$, high plasma levels of the TNF$\alpha$ protein were associated with an improved prognosis for patients in the total cohort and in Group 1 but not in Group 2. Patients that expressed the AA genotype for the TNF$\alpha$-SNP did not show a significant difference in BCSS compared to patients with the GG genotype (total population $p$ = 0.064) and there was no correlation between the TNF$\alpha$ protein levels and TNF-$\alpha$-SNP genotype ($R^2$ = 0.028, $p$ = 0.45).

**Table 4.** Cytokine gene polymorphisms and clinical outcome for patients treated with high-dose chemotherapy including cyclophosphamide (Group 1) or paclitaxel (Group 2).

| | Breast Cancer Specific Survival | | | | | | | | | | | | | | | |
| | Total Population | | | | | Group I | | | | | Group 2 | | | | | |
| | N | OS | | | | N | OS | | | | N | OS | | | | |
| Variable | 130 | Median Survival (Months) | Hazard Ratio (95% CI) | $X^2$ | *p* | 74 | Median Survival (Months) | Hazard Ratio (95% CI) | $X^2$ | *p* | 56 | Median Survival (Months) | Hazard Ratio (95% CI) | $X^2$ | *p* |
|---|---|---|---|---|---|---|---|---|---|---|---|---|---|---|---|
| ILRA SNP 1 | 117 | | | | | 53 | | | | | 53 | | | | |
| CC | 66 | 15.3 | 1 (reference) | | | 33 | 14.3 | 1 (reference) | | | 32 | 17.9 | 1 (reference) | | |
| CT | 45 | 26.6 | 1.60 (1.10–2.35) | 6 | 0.015 | 26 | 28.5 | 2.22 (1.28–3.84) | 8.1 | 0.004 | 20 | 25.1 | 1.33 (0.76–2.32) | 0.98 | 0.32 |
| TT | 6 | 12.8 | 0.40 (0.13–1.25) | 2.5 | 0.11 | 5 | 9.5 | 0.46 (0.14–1.55) | 1.6 | 0.21 | 1 | 16 | 0.57 (0.004–6.84) | 0.2 | 0.65 |
| CC | 66 | 15.3 | 1 (reference) | | | 33 | 14.3 | 1 (reference) | | | | 17.9 | 1 (reference) | | |
| CT + TT | 51 | 25.3 | 1.46 (1.01–2.12) | 4.1 | 0.043 | 31 | 26.6 | 1.87 (1.11–3.17) | 5.5 | 0.019 | | 24.9 | 1.29 (0.74–2.24) | 0.82 | 0.37 |
| ILRA SNP 2 | 116 | | | | | 64 | | | | | 53 | | | | |
| TT | 47 | 21.4 | 1 (reference) | | | 31 | 19.2 | 1 (reference) | | | 17 | 27.5 | 1 (reference) | | |
| CT | 50 | 20.1 | 1.06 (0.70–1.58) | 0.067 | 0.8 | 24 | 25.2 | 1.06 (0.61–1.83) | 0.042 | 0.84 | 26 | 17.1 | 1.09 (0.58–2.05) | 0.74 | 0.79 |
| CC | 19 | 14.4 | 1.44 (0.80–2.58) | 1.48 | 0.5 | 9 | 14.4 | 2.05 (0.82–5.14) | 2.3 | 0.13 | 10 | 20.4 | 1.11 (0.49–2.49) | 0.058 | 0.81 |
| TT | 47 | 21.4 | 1 (reference) | | | 31 | 19.2 | 1 (reference) | | | 17 | 27.5 | 1 (reference) | | |
| CT+CC | 69 | 18.6 | 1.14 (0.78–1.65) | 0.44 | 0.5 | 33 | 19.9 | 1.21 (0.73–2.00) | 0.53 | 0.46 | 36 | 18.8 | 1.09 (0.61–1.19) | 0.81 | 0.78 |
| TNF SNP | 89 | | | | | 57 | | | | | 32 | | | | |
| GG | 50 | 14.4 | 1 (reference) | | | 27 | 14.3 | 1 (reference) | | | 23 | 15.1 | 1 (reference) | | |
| AG | 36 | 18.1 | 0.67 (0.43–1.04) | 3.1 | 0.076 | 27 | 19.2 | 0.59 (0.33–1.05) | 3.2 | 0.074 | 9 | 16 | 1.08 (0.48–2.41) | 0.36 | 0.85 |
| AA | 3 | 31.9 | 0.71 (0.26–1.93) | 0.45 | 0.5 | 3 | 31.9 | 0.70 (0.24–1.99) | 0.45 | 0.5 | 0 | | | | |
| GG | 50 | 14.4 | 1 (reference) | | | 27 | 14.3 | 1 (reference) | | | | | | | |
| GG+AG | 39 | 18.5 | 0.67 (0.43–1.02) | 4.4 | 0.064 | 30 | 19.5 | 0.59 (0.33–1.03) | 3.5 | 0.063 | | | | | |
| IL-6 SNP | 117 | | | | | 65 | | | | | 52 | | | | |
| GG | 47 | 22.9 | 1 (reference) | | | 27 | 24.8 | 1 (reference) | | | 20 | 20.8 | 1 (reference) | | |
| CG | 53 | 19.2 | 0.89 (0.60–34) | 0.29 | 0.58 | 29 | 19.2 | 0.99 (0.58–1.68) | 0.002 | 0.96 | 24 | 20.4 | 0.77 (0.41–1.44) | 0.68 | 0.41 |
| CC | 17 | 15.8 | 0.90 (0.51–1.58) | 0.13 | 0.71 | 9 | 13.4 | 1.00 (0.45–2.23) | 8E-05 | 0.99 | 8 | 27.8 | 0.78 (0.34–1.77) | 0.36 | 0.55 |
| GG | 47 | 22.9 | 1 (reference) | | | 27 | 24.8 | 1 (reference) | | | 20 | 20.8 | 1 (reference) | | |

**Table 4.** *Cont.*

| | Breast Cancer Specific Survival | | | | | | | | | | | | | | | |
| --- | --- | --- | --- | --- | --- | --- | --- | --- | --- | --- | --- | --- | --- | --- | --- | --- |
| | Total Population | | | | | Group I | | | | | Group 2 | | | | | |
| | N | | OS | | | N | | OS | | | N | | OS | | | |
| Variable | 130 | Median Survival (Months) | Hazard Ratio (95% CI) | X² | *p* | 74 | Median Survival (Months) | Hazard Ratio (95% CI) | X² | *p* | 56 | Median Survival (Months) | Hazard Ratio (95% CI) | X² | *p* |
| CG + CC | 70 | 18.9 | 0.89 (0.60–1.29) | 0.39 | 0.53 | 38 | 17.5 | 0.97 (0.59–1.59) | 0.19 | 0.89 | 32 | 22.9 | 0.77 (0.42–1.39) | 0.76 | 0.38 |
| IL-1B SNP1 | 117 | | | | | 64 | | | | | 53 | | | | |
| CC | 54 | 15.9 | 1 (reference) | | | 25 | 15 | 1 (reference) | | | 29 | 16.3 | 1 (reference) | | |
| CT | 50 | 25.2 | 1.44 (0.97–2.14) | 3.3 | 0.07 | 31 | 25.7 | 1.68 (0.95–2.97) | 3.2 | 0.073 | 19 | 24.9 | 1.22 (0.68–2.18) | 0.45 | 0.5 |
| TT | 13 | 24.8 | 1.10 (0.60–2.00) | 0.087 | 0.77 | 8 | 25.7 | 1.33 (0.61–2.88) | 0.52 | 0.47 | 5 | 17.3 | 0.95 (0.36–2.53) | 0.009 | 0.93 |
| CC | 54 | 15.9 | 1 (reference) | | | 25 | 15 | 1 (reference) | | | 29 | 16.3 | 1 (reference) | | |
| CT + TT | 63 | 25.1 | 1.38 (0.93–2.00) | 2.7 | 0.098 | 39 | 25.7 | 1.64 (0.94–2.85) | 3.1 | 0.078 | 24 | 23.9 | 1.16 (0.67–2.01) | 0.28 | 0.6 |
| IL-1B SNP2 | 117 | | | | | 62 | | | | | 53 | | | | |
| CC | 66 | 19.9 | 1 (reference) | | | 37 | 21.4 | 1 (reference) | | | 29 | 18.5 | 1 (reference) | | |
| CT | 45 | 21.4 | 0.83 (0.33–2.06) | 0.16 | 0.69 | 24 | 17.1 | 0.89 (0.53–1.51) | 0.18 | 0.67 | 21 | 24.1 | 1.11 (0.63–1.96) | 0.13 | 0.73 |
| TT | 6 | 14.6 | 0.99 (0.67–1.45) | 0.006 | 0.94 | 3 | 15 | 1.32 (0.44–3.85) | 0.25 | 0.62 | 3 | 14.1 | 0.37 (0.072–1.91) | 1.4 | 0.24 |
| CC | 66 | 19.9 | 1 (reference) | | | 37 | 21.4 | 1 (reference) | | | 29 | 18.5 | 1 (reference) | | |
| CT + TT | 51 | 19.9 | 0.97 (0.67–1.40) | 0.03 | 0.86 | 27 | 16.7 | 0.94 (0.56–1.56) | 0.067 | 0.8 | 24 | 23.1 | 1.03 (0.59–1.78) | 0.009 | 0.92 |

Text in red indicates a significant difference based on biomarker

.

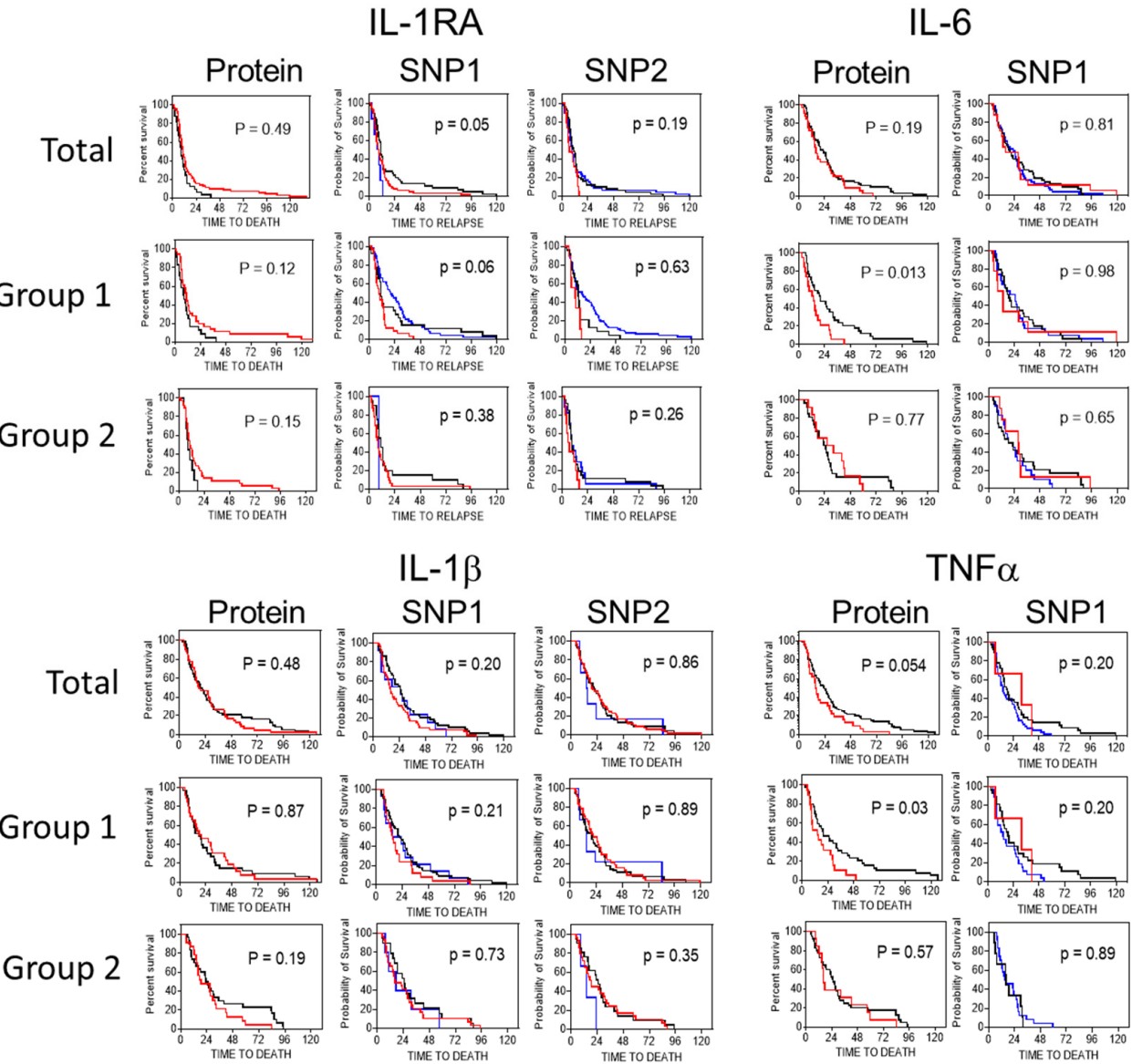

**Figure 3.** Comparison of cytokine levels and SNPs on clinical outcome. The impact of the relative levels of plasma cytokines (protein) or specific SNP genotypes (SNP) on breast-cancer-specific survival for all of the patients (total) or patients treated using cyclophosphamide- (Group 1) or paclitaxel-based (Group 2) high-dose chemotherapy was determined. The patients were characterized for high levels of plasma cytokine or "risk" cytokine SNP (blue line) or low levels or SNP (red line) for the indicated cytokines and the percent survival analyzed using Kaplan–Meier survival curves. The indicated *p* values were determined using log-rank statistics for the comparison.

### 3.5. Combination of Variant Genotypes and Survival

We explored the effect of the number of variant genotypes (IL-1RA-SNP-01 CC, IL-1RA-SNP-02 CC, IL-1β-SNP-01 CC, IL-1β-SNP-02, IL-6-SNP AA, and TNFα-SNP AA) on survival even though only IL-1RA-SNP-01 was significant on its own (Table 5). The median BCSS decreased progressively for the total cohort of patients carrying 2–5 "risk" genotypes with a significant log rank test for trend ($p = 0.0006$). The median difference in BCSS for 2 variant genotype SNPs was 18.1 vs. 9.64 months, HR = 0.40 (95% CI = 0.18–0.94, $p = 0.034$) and for 5 variant alleles the difference in BCSS was 37.6 vs. 15.4 months, HR = 0.44 (95% CI = 0.25–0.77, $p = 0.0040$). For patients in Group 1, the increase in the number of risk alleles had a strong trend for decreased BCSS ($p = 0.0040$). Alternately, the effect of combining putative "negative" plasma levels for a number of cytokines together increased the predictive value even when, individually,

these cytokine levels did not indicate a significant difference in outcome. There was a significant decreased trend in median BCSS as the number of "negative" plasma cytokines was increased for the total group of patients ($p = 0.012$) or patients in Group 1 ($p = 0.020$) but not for patients in Group 2. The difference in BCCS for patients with any two "negative" cytokines was 23.1 vs. 15.1 months, HR = 0.62 (95% CI = 0.36–1.06, $p = 0.082$) while for a combination of three elevated cytokines, the difference was 28.1 vs. 17.7 months, HR = 0.47 (95% CI = 0.30–0.74, $p = 0.0013$). The combination of the number of "risk" genotypes and "negative" plasma levels also showed a decreased trend in BCSS for the total group of patients ($p = 0.0051$) and patients in Group 1 (0.0026) but not for patients in Group 2.

**Table 5.** Effect of combining risk conditions on clinical outcome for patients treated with high-dose chemotherapy.

| Group | Measure | Outcome | Survial for the Number of "Risk" Measures of Cytokine | | | | | | | | Logrank Test for Trend | |
| | | | 0 | 1 | 2 | 3 | 4 | 5 | 6 | 7 | chi$^2$ | *p* |
|---|---|---|---|---|---|---|---|---|---|---|---|---|
| Total | Plasma | PFS | 11.5 | 11.1 | 8.7 | 9.7 | 7.6 | 2 | | | 6.9 | 0.0088 |
| | | BCSS | 28.9 | 19.9 | 13.8 | 13.8 | 23.3 | 5.8 | | | 6.2 | 0.012 |
| | SNP | PFS | 10.8 | 11.6 | 8.6 | 6.9 | 6.2 | 2 | | | 8.8 | 0.003 |
| | | BCSS | 28.4 | 25.3 | 16.1 | 12.1 | 10.4 | 13.4 | | | 11.8 | 0.0006 |
| | Plasma+SNP | PFS | 19.9 | 10.4 | 9.1 | 10.4 | 8.3 | 9.7 | 6.6 | 5.5 | 4.7 | 0.031 |
| | | BCSS | 49 | 25.1 | 19.2 | 22.9 | 14.4 | 20.9 | 12.1 | 9.6 | 7.8 | 0.0051 |
| Group 1 | Plasma | PFS | 9.5 | 11.6 | 4.7 | 6.9 | 7.2 | 2 | | | 9.8 | 0.0001 |
| | | BCCS | 19.6 | 19.9 | 9.5 | 12.1 | 18.8 | 5.8 | | | 5.4 | 0.02 |
| | SNP | PFS | 10.8 | 11.3 | 8.7 | 6.7 | 4.7 | | | | 6.1 | 0.013 |
| | | BCSS | 28.4 | 26.1 | 17.5 | 10.8 | 7.9 | | | | 8.3 | 0.004 |
| | Plasma+SNP | PFS | 27.3 | 10.1 | 9.1 | 12.3 | 9.5 | 4.7 | 6.2 | 5 | 9.2 | 0.0025 |
| | | BCSS | 66.9 | 22.5 | 19.2 | 24.8 | 19.6 | 6.8 | 10.4 | 9.2 | 9 | 0.0026 |
| Group 2 | Plasma | PFS | 12.6 | 8.5 | 10.6 | 12.9 | 10.2 | - | | | 0.05 | 0.82 |
| | | BCSS | 30.4 | 18.6 | 14.4 | 15.4 | 29.5 | - | | | 0.75 | 0.39 |
| | SNP | PFS | 9.2 | 1.9 | 6.8 | 8.9 | 7.6 | 14.1 | | | 0.19 | 0.67 |
| | | BCSS | 16 | 22.9 | 12.4 | 12.3 | 17.7 | 14.1 | | | 0.7 | 0.4 |
| | Plasma+SNP | PFS | 12.4 | 12.8 | 10.5 | | 8.3 | 10.6 | 16 | 14.1 | 1 | 0.32 |
| | | BCSS | 31.1 | 34.9 | 20.7 | | 14.4 | 29.4 | 41.8 | 14.1 | 0.00008 | 0.99 |

Text in red indicates a significant difference for trend.

### 3.6. Cytokine Levels and Clinicopathological Characteristics

Hormone receptor status (estrogen receptor or progesterone receptor) did not correlate with PFS and BCSS for this population but HER-2 status indicated significant differences in PFS and BCSS for the total cohort and patients in Group 1 and Group 2 (Supplementary Table S3). There were significant differences in PFS and BCSS for patients with more than one metastatic site and for metastatic site(s) that included liver and lung for the total cohort as well as for patients in both Group 1 and Group 2. The results of tests of association between IL-1RA, IL-1β, IL-6, IL-2, and TNFα positive status and clinicopathological characteristics showed no association between cytokine marker expression and age, sHER-2 expression, estrogen or progesterone receptor status, previous hormone treatment, or the number of documented sites of metastases Table 6. However, there were significant associations between IL-1RA levels and bone and lung metastasis, IL-1β and lung metastasis, and IL-2 and liver metastasis.

**Table 6.** Correlation of plasma cytokine levels with patient clinical characteristics.

| | | IL-RA1 | | | IL-1b | | | TNFa | | | IL-6 | | | IL-2 | | |
|---|---|---|---|---|---|---|---|---|---|---|---|---|---|---|---|---|
| | N = 130 | pos | % | *p* | pos | % | *p* | pos | % | *p* | pos | % | *p* | pos | % | *p* |
| **Age** | | | | | | | | | | | | | | | | |
| <40 | 32 | 18 | 75 | 0.67 | 11 | 42 | 0.53 | 10 | 39 | 0.77 | 6 | 30 | 0.75 | 11 | 48 | 0.81 |
| 40–49 | 66 | 34 | 64 | | 25 | 43 | | 16 | 28 | | 18 | 37 | | 20 | 40 | |
| 50–59 | 32 | 19 | 70 | | 14 | 48 | | 7 | 24 | | 7 | 30 | | 10 | 37 | |
| **ER** | | | | | | | | | | | | | | | | |
| Negative | 47 | 25 | 69 | 0.86 | 15 | 36 | 0.44 | 13 | 32 | 0.32 | 11 | 32 | 0.93 | 16 | 43 | 0.67 |
| Positive | 66 | 34 | 67 | | 26 | 47 | | 13 | 24 | | 15 | 33 | | 20 | 40 | |
| **PR** | | | | | | | | | | | | | | | | |
| Negative | 51 | 28 | 68 | 0.96 | 16 | 40 | 0.43 | 13 | 29 | 0.59 | 14 | 39 | 0.22 | 13 | 37 | 0.28 |
| Positive | 57 | 31 | 70 | | 22 | 47 | | 12 | 26 | | 10 | 26 | | 20 | 47 | |
| **HER-2 (tissue or sHER-2)** | | | | | | | | | | | | | | | | |
| Negative | 63 | 38 | 69 | 0.75 | 24 | 41 | 0.37 | 16 | 27 | 0.61 | 15 | 30 | 0.48 | 21 | 39 | 0.83 |
| Positive | 54 | 29 | 64 | | 25 | 50 | | 16 | 32 | | 16 | 38 | | 19 | 44 | |
| **Adjuvant endocrine therapy** | | | | | | | | | | | | | | | | |
| No | 83 | 50 | 71 | 0.11 | 34 | 45 | 0.31 | 23 | 30 | 0.54 | 26 | 41 | <span style="color:red">0.001</span> | 26 | 39 | 0.75 |
| Yes | 44 | 20 | 61 | | 14 | 48 | | 10 | 28 | | 5 | 18 | | 15 | 45 | |
| **Documented Sites of Metastases** | | | | | | | | | | | | | | | | |
| NED, 1 | 75 | 39 | 65 | 0.48 | 27 | 42 | 0.58 | 15 | 23 | 0.11 | 17 | 31 | 0.71 | 23 | 38 | 0.85 |
| ≥ 2 | 55 | 32 | 73 | | 23 | 47 | | 18 | 38 | | 14 | 37 | | 18 | 45 | |
| **Bone Metastases** | | | | | | | | | | | | | | | | |
| No | 71 | 53 | 72 | <span style="color:red">>0.001</span> | 34 | 42 | 0.25 | 24 | 30 | 0.14 | 22 | 33 | 0.28 | 28 | 37 | 0.32 |
| Yes | 41 | 18 | 58 | | 16 | 50 | | 9 | 29 | | 9 | 36 | | 13 | 45 | |
| **Lung Metastases** | | | | | | | | | | | | | | | | |
| No | 89 | 47 | 66 | <span style="color:red">0.02</span> | 32 | 42 | <span style="color:red">0.04</span> | 20 | 26 | 0.06 | 23 | 37 | 1.00 | 28 | 41 | 0.29 |
| Yes | 31 | 24 | 73 | | 18 | 49 | | 13 | 36 | | 8 | 28 | | 13 | 42 | |
| **Liver Metastases** | | | | | | | | | | | | | | | | |
| No | 109 | 56 | 65 | 0.24 | 38 | 40 | 0.08 | 25 | 27 | 0.17 | 26 | 34 | 1.00 | 30 | 37 | <span style="color:red">0.04</span> |
| Yes | 21 | 14 | 78 | | 12 | 63 | | 8 | 42 | | 5 | 33 | | 11 | 61 | |
| **Lymph Node** | | | | | | | | | | | | | | | | |
| No | 89 | 48 | 70 | 0.85 | 33 | 45 | 0.71 | 21 | 29 | 0.67 | 19 | 31 | 0.38 | 28 | 42 | 0.95 |
| Yes | 42 | 24 | 67 | | 17 | 44 | | 12 | 31 | | 12 | 40 | | 13 | 38 | |
| **Other sites** | | | | | | | | | | | | | | | | |
| No | 104 | 55 | 66 | 0.51 | 41 | 45 | 0.82 | 29 | 33 | 0.48 | 24 | 32 | 0.80 | 34 | 41 | 0.64 |
| Yes | 26 | 16 | 76 | | 9 | 41 | | 9 | 41 | | 7 | 39 | | 7 | 39 | |

Text in red indicates a significant association with the clinicopathologic feature.

## 4. Discussion

The results from this study showed that plasma levels of IL-1RA, TNF$\alpha$, and IL-6 and a specific SNP in the IL-1RA gene were associated with clinical outcome for patients with metastatic breast cancer treated with HDC. The ability of cytokine levels or genetic polymorphisms to indicate clinical outcome depended on the chemotherapy regimen

used to treat the patients. Patients included in this study were treated with high-dose cyclophosphamide- (Group 1) or high-dose paclitaxel-containing chemotherapy (Group 2). Except for their treatment regimens, the patients in Group 1 and Group 2 were very similar and were matched for age, hormone receptor status, and clinical outcome following treatment. In addition, the levels of the plasma cytokines and the frequency of the different SNPs were similar between the two groups of patients.

Higher levels of IL-1RA protein in plasma were associated with an improved response to therapy for patients that were treated with high-dose cyclophosphamide-containing chemotherapy (Group1) but not for patients treated with high-dose paclitaxel-containing chemotherapy (Group 2). Higher levels of TNFα and IL-6 were associated with decreased BCSS only for patients in Group 1. Patients in Group 2 could not be distinguished by the levels of TNFα or IL-6. The level of IL-1β in patient plasma did not indicate a difference in clinical outcome for any of the groups of patients. The cytokine levels used to determine biomarker status were determined in samples taken prior to treatment and represents baseline levels. Because all of the patients had advanced metastatic disease it is unlikely that the difference in cytokine levels results from an association with metastasis as previously reported [50]. Therefore, the difference in the ability of different levels of cytokines to predict outcome may be dependent on treatment with a particular chemotherapy regimen. We have previously shown that the ability of secreted ICAM-1 and VCAM-1 to indicate differences in outcome also depend on the particular type of chemotherapy in this patient population: the levels of these secreted adhesion molecule fragments correlated with prognosis only for patients treated with high-dose cyclophosphamide-containing chemotherapy [51]. However, the levels of secreted ICAM-1 and VCAM-1 did not correlate with the levels of any of the cytokines tested in this study.

The IL-1RA-SNP-01 genotype was able to indicate differences in clinical outcome for patients treated with high-dose cyclophosphamide-containing chemotherapy while the IL-1RA-SNP-02 genotype could not. The IL-1RA-SNP-01 genotype has been previously shown to correlate with the amount of IL-1RA protein secreted into tissue and plasma [38] which would be consistent with this result. However, the IL-1RA-SNP-01 genotype and the level of IL-1RA in plasma were only weakly correlated in this population. The TNFα SNP genotype has also been previously shown to correlate with the level of protein expressed [41]. The TNFα-SNP was able to indicate differences in outcome for PFS in patients in Group 1 and a high level of plasma TNFα was able to indicate a poorer outcome for both PFS and BCSS for patients in Group 1 (but not for patients in Group 2). However, in this patient population, the SNP genotype and plasma levels of cytokine were not correlated. A high level of plasma IL-6 protein correlated with a poorer PFS and BCSS for patients in Group 1 but the IL-6 SNP genotype did not correlate with outcome for these patients. The IL-6 SNP genotype and IL-6 plasma protein levels also did not correlate in this population of patients although this SNP has previously been reported to correlate with protein expression [39]. The lack of correlation between the SNPs and the level of plasma cytokines might be related to the advanced metastatic disease in these patients since cytokine expression has been shown to be elevated in patients with metastasis [50] and may indicate why plasma cytokine levels may be a better predictor of outcome. In addition, changes in the levels of microRNA that can post-transcriptionally affect the level of cytokine proteins have been associated with cancer progression [52] and may disrupt correlation between SNPs and plasma levels of cytokine. The IL-1β SNPs and level of plasma IL-1β were not informative for clinical outcome for this group of patients.

These results showed a weak correlation between plasma cytokine levels and cytokine genotype and showed that the level of plasma protein was informative more often for these patients with metastatic breast cancer treated with HDC. While there were only weak correlations between a cytokine SNP and its plasma cytokine level, the levels of circulating cytokines were correlated with each other. For example, the level of TNFα significantly correlated with the levels of IL-1β, IL-2, and IL-6 with a Pearson R of between 0.6 and 0.8. This indicates that patients are more likely to have simultaneously elevated

levels of multiple cytokines, possibly as a response to having metastatic disease, rather than having an elevated level of a specific cytokine in response to a particular SNP. This suggests that systemic or environmental stimuli have a larger impact on the plasma levels of cytokines than do differences in gene sequence in patients with metastatic breast cancer. While many studies have found associations between different SNPs in cytokine genes and carcinogenesis or prognosis for patients with breast cancer, our data suggest that for patients with metastatic cancers these differences were less useful than measures of baseline cytokine levels. Increasing the number of "risk" alleles or "negative" cytokine levels improved the prognostic power of the comparison even when these characteristics were not significant individually. This suggests that assignment of risk may involve a large number of candidates that can contribute even when at sub-threshold levels. Furthermore, there were significant differences in the ability of both cytokine SNPs and the baseline cytokine protein levels to indicate outcome which are dependent on the particular type of chemotherapy used to treat the patients.

It has been shown that different chemotherapy drugs have differential effects on the activation of an inflammatory response [53]. For example, treatment of breast cancer patients with paclitaxel has been shown to enhance the production of multiple cytokines including IL-6, IL-8, and IL-10 and produce fatigue and joint pain [54]. In vitro, treatment of monocytes, T cells, or breast cancer cells with paclitaxel can increase levels of the proinflammatory cytokines, IL-1β and TNFα [55]. Treatment with paclitaxel can also indirectly promote inflammatory processes by selectively inhibiting suppressor myeloid cells and suppressor T cells which could potentially enhance immune responses in paclitaxel-treated patients [55,56]. In contrast, cyclophosphamide treatment is often used to eliminate subsequent immune responses by decreasing the number of both cytotoxic T cells and T helper cells [57]. High doses of cyclophosphamide are used for immunoablation to completely block immune responses [53]. Furthermore, treatment of breast cancer patients with cyclophosphamide-, methotrexate-, and fluorouracil-containing chemotherapy has been linked to small decreases in plasma IL-6 and IL-8 [58] in studies of chemotherapy-induced cognitive decline. Therefore, the different immune responses to the different chemotherapy treatments might be sufficient to disrupt the effects expected from baseline measures of cytokine levels or cytokine genetic polymorphisms [59]. For example, paclitaxel treatment might elevate proinflammatory cytokines during treatment to mask the potential effects of baseline levels on clinical outcome while innate immune responses are less affected in cyclophosphamide-treated patients.

Since changes in cytokine levels have been reported in response to treatment with some chemotherapy regimens, we examined whether these changes could indicate prognosis. A subgroup of the patients treated with cyclophosphamide had samples staged at baseline, during treatment, and after treatment. The median levels of the cytokines were not different among the three time points for all of the patients indicating that changes in cytokine levels in response to treatment were not common for this group of patients. We then examined whether changes in cytokines during treatment for each patient correlated with clinical outcome by determining if the level of cytokine increased between baseline and during chemotherapy, between chemotherapy and after treatment, and between baseline and after treatment. A change in IL-1RA, IL-1β, IL-6, and TNFα between baseline and chemotherapy did not indicate a difference in clinical response. However, an increase in IL-1β (and IL-1RA) levels between chemotherapy and after treatment was weakly associated with a poorer outcome. An increase in the amount of IL-2 between baseline and chemotherapy and between baseline and the end of treatment was also associated with a poorer outcome for these patients.

This group of patients has previously been examined for a variety of plasma and genetic biomarkers for clinical outcome. Our previous results have shown a significant contribution from elevated levels of secreted HER-2 [46,47] and sFas [60] to predicting a poorer outcome indicating contributions from growth factor or apoptosis pathways. Both sHER-2 and sFas levels were able to indicate prognosis for patients in Group 1 and

Group 2. The responses to chemotherapy as indicated by the ability of SNPs in DNA repair [61] and detoxification enzyme genes [62] could also predict clinical outcome for patients in both Group 1 and Group 2. However, our studies looking at differences in immune system responses were able to indicate clinical responses only for patients in Group 1. We previously showed that the secreted levels of the inflammatory adhesion molecules VCAM-1 and ICAM-1 can indicate the outcome for patients in Group 1 but not in Group 2 [51]. In the current study, we showed that cytokine levels and SNPs can indicate outcome for patients in Group 1 but not in Group 2. It is interesting to note that the ability of the inflammatory mediators and cytokines to indicate prognosis is strongly dependent on the chemotherapy regimen used to treat the patients while the tumor-derived markers such as HER-2, or the detoxification and repair pathways, do not appear to be strongly dependent on the chemotherapy regimen. This suggests that the immune and inflammatory biomarkers interact differently with the chemotherapy regimen than the tumor- or genetic-based biomarkers. However, while the expression levels of different cytokines appear to be correlated with one another (elevated levels of TNF$\alpha$ correlate with elevated levels of IL-1RA, IL-1$\alpha$, IL-2, and IL-6), they do not correlate with elevated levels of ICAM-1 and VCAM-1, suggesting the levels of these biomarkers differ in their response to cancer progression. The recent development of immune modulators, such as the "checkpoint" inhibitors, that activate immune responses against tumors have shown a strong impact on treatment of malignant melanoma and are starting to be used for treatment of patients with breast cancer. Our results indicate that the impact of baseline levels of cytokines, or on the ability of cytokine SNPs, to indicate clinical outcome is dependent on the chemotherapy regimen. This suggests that different chemotherapy regimens might be more effective in patients also treated with checkpoint inhibitors.

## 5. Conclusions

The plasma levels of IL-1RA, IL-1$\beta$, or TNF$\alpha$ or specific SNPs in IL1RA can indicate clinical outcome for some groups of patients with metastatic breast cancer being treated with chemotherapy. The utility of using cytokines or cytokine SNPs as a biomarker for clinical outcome for patients with metastatic breast cancer depends on the chemotherapy regiment used for treatment. For example, different chemotherapy regimens can differentially affect the ability of cytokine levels or SNPs to predict outcome in patients with metastatic breast cancer and could differentially affect the potential success of immune therapies.

**Supplementary Materials:** The following supporting information can be downloaded at: https://www.mdpi.com/article/10.3390/immuno3010002/s1, Table S1: Plasma cytokine levels and determination of cutpoint concentrations; Table S2: Identification of DNA probes used to detect cytokine SNPs; Table S3: Patient clinicopathological characteristics and clinical outcome.

**Author Contributions:** Conceptualization, R.L., M.B. and M.C.; methodology, R.L., M.B. and M.C.; validation, R.L., M.B., C.B. and M.C.; formal analysis, R.L. and M.B.; resources, R.L.; data curation, R.L. and M.C.; writing—original draft preparation, R.L.; writing—review and editing, R.L., M.B., C.B. and M.C.; supervision, R.L.; project administration, M.B.; funding acquisition, R.L. All authors have read and agreed to the published version of the manuscript.

**Funding:** This research was partially funded by the Northern Cancer Research Foundation, Sudbury, Ontario, P3E 5K1.

**Institutional Review Board Statement:** The study was conducted in accordance with the Declaration of Helsinki, and approved by the Institutional Review Board of Laurentian Hospital, Sudbury, Ontario.

**Informed Consent Statement:** Informed consent was obtained from all subjects involved in the study.

**Data Availability Statement:** The data presented in this study are available on request from the corresponding author.

**Conflicts of Interest:** The authors declare no conflict of interest. The funders had no role in the design of the study; in the collection, analyses, or interpretation of data; in the writing of the manuscript; or in the decision to publish the results.

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
