# Peer review of "Plasma Cytokine Levels and Cytokine Genetic Polymorphisms in Patients with Metastatic Breast Cancer Receiving High-Dose Chemotherapy"

_2673-5601, doi:10.3390/immuno3010002_

Round 1

Reviewer 1 Report

The authors investigated the plasma cytokine levels and cytokine gene SNPs in patients with metastatic breast cancers treated with high-dose cyclophosphamide or paclitaxel. Here are a few additional minor suggestions.

1 Serum cytokines can predict outcome for patients in the Group1, not Group 2. Cyclophosphamide and paclitaxel are two commonly used chemotherapeutic agents in cancer treatments, which may induce different immune responses due to their mechanism of action. In addition, IL-1beta, TNF-É‘ and IL-6 are inflammation cytokines, and IL-1RA is an anti-inflammatory cytokine, which coordinately regulate innate immunity and outcome. If the authors would cite more related references about these researches, which can strengthen the rationale and make the description clearer.

2. Higher levels of specific cytokines may be induced by metastases. For examples, IL-1 beta is a pro-inflammatory cytokine whose expression in primary tumors with bone metastasis. IL-6 produced in the bone marrow microenvironment to bone metastasis.  To strengthen the rationale, it would be better to discuss it.

3. One limitation of this study is that the author identified genetic association of TNFA/IL6 gene with only one SNP and did not find the correlation between genotypes and protein serum level of these two cytokines. The authors should classify or discuss it.

4. There is a typographical error in the line 497. Group 249 should be Group 2.

Author Response

Thanks for the comments.  In this revision, I have added a few more references and some comments related to the role of inflammation in cancer progression and how that might be relevant to the observed prognostic effect of cytokines biomarkers. 

I have also indicated that the presence of metastasis might be relevant to the level of cytokines measured in the patients and how this might impact the correlation between cytokine SNP and plasma cytokine levels.

I have also gone through the manuscript to repair typographical errors and to clarify a few statements.

Reviewer 2 Report

In this article the authors showed serum levels of IL-1RA, TNF alpha, and IL-6 and a specific SNP in the IL-1RA gene are associated with clinical outcome for patients with 389 metastatic breast cancer treated with HDC. The article is a complete statistical processing of patient samples from 1991-1997.

The article is well written and the results support the steatments.

Author Response

Thank you very much for your comments.  I have made a few changes to the manuscript to fix some typographical errors and clarify a few statements.

Reviewer 3 Report

Robert Lafrenie and colleagues described in this manuscript the levels of plasma cytokine and cytokine genetic polymorphisms in metastatic breast cancer patients receiving high-dose chemotherapy. They analyzed the plasma levels and characterized individual nucleotide polymorphisms for IL-1RA, IL-1b, IL-2, IL-6 and TNFa in 130 metastatic breast cancer patients (Group 1, 74 patients treated with high dose cyclophosphamide & Group 2, 56 patients treated with high dose paclitaxel-containing regimens). Their endpoint conclusion is that the immune cytokines may be useful as prognostic biomarkers in the treatment of metastatic breast cancer patients treated with different types of high-dose chemotherapy. The experiments & analysis are well performed, and the conclusions are appropriate. Although the manuscript generally presented in a logical order, the text should be revised to improve clarity of the writing. I believe after substantial input from the authors in this manuscript may considered for publication in Immuno Journal, MDPI. The authors should consider the following issues to improve the strength of this manuscript:  

·      I think authors mixed up throughout the manuscript the levels of cytokines in plasma or serum. They should critically revise and give attention to this minor but a major mistake as the serum and plasma composition is not the same. 

·      Figures quality should be improved (the figures in the illustration look blurry).

·      I noticed in some sentences (such as line 14) after “cyclophosphamide”, unwanted hyphen used. Authors need to be revised throughout the manuscript to fix these types of small errors.  

·      I would also like to suggest the authors to include “List of Abbreviations”. 

·      I recommend the authors to use the correct format of references in this manuscript according to this Journal’s guidelines. 

Author Response

Thank you very much for the constructive comments.  

We have clarified the manuscript to show that we used plasma preparations for cytokine analysis and have therefore consistently referred to plasma in the description of our results. (Some reports in the introduction did use serum levels of cytokines which were not changed in the manuscript.)

The figures have been replaced with higher resolution images and I hope this is maintained in the transfer of files.

I have made several changes to fix typographical and other errors and have tried to make some statements more clear.   I hope that this has improved the readability of the manuscript. 

I have carefully followed the reference format included in the manuscript template - although I have included the doi numbers for all relevant references.  

I did not include a list of abbreviations, but have removed a few abbreviations that were only used a few times.  What is left is HDC (high dose chemotherapy), ASCT (autologous stem cell transplant), BCSS (breast cancer specific survival), PFS, and SNP (single nucleotide polymorphism).